# RNAi Screen Identifies AXL Inhibition Combined with Cannabinoid WIN55212-2 as a Potential Strategy for Cancer Treatment

**DOI:** 10.3390/ph17111465

**Published:** 2024-11-01

**Authors:** Feifei Li, Hang Gong, Xinfei Jia, Chang Gao, Peng Jia, Xin Zhao, Wenxia Chen, Lili Wang, Nina Xue

**Affiliations:** 1Institute of Pharmacology and Toxicology, Academy of Military Medical Sciences, Academy of Military Sciences, Beijing 100850, China; fei-leaves@163.com (F.L.); 15210142562@163.com (X.J.); jiapeng.2010@163.com (P.J.); 2State Key Laboratory of Bioactive Substances and Functions of Natural Medicines, Institute of Materia Medica, Chinese Academy of Medical Sciences and Peking Union Medical College, Beijing 100050, China; gonghang@imm.ac.cn (H.G.); gaochang@imm.ac.cn (C.G.); zhaoxin@imm.ac.cn (X.Z.); wenxiachen@imm.ac.cn (W.C.); 3Beijing Key Laboratory of Non-Clinical Drug Metabolism and PK/PD Study, Institute of Materia Medica, Chinese Academy of Medical Sciences and Peking Union Medical College, Beijing 100050, China

**Keywords:** RNAi screening, AXL, cannabinoid, combination chemotherapy, cancer, cytotoxic T cells

## Abstract

**Background and objective**: Cannabinoids are commonly used as adjuvant cancer drugs to overcome numerous adverse side effects for patients. The aim of this study was to identify the target genes that show a synergistic anti-tumor role in combination with the cannabinoid WIN55212-2 in vitro and in vivo. **Methods**: A human kinome RNAi library was used to screen the targeted gene that silencing plus WIN55212-2 treatment synergistically inhibited cancer cell growth in an INCELL Analyzer 2000. Cell viability, cell phase arrest and apoptosis were evaluated by MTT and flow cytometry assay. In vivo combined anti-tumor effects and regulatory mechanisms were detected in immunocompromised and immunocompetent mice. **Results**: Using RNAi screening, we identified the tyrosine receptor kinase AXL as a potential gene whose silencing plus WIN55212-2 treatment synergistically inhibited the proliferation of cancer cells in an INCELL Analyzer 2000. Subsequently, we demonstrated that inhibition of AXL by TP-0903 potentiated the inhibitory role of WIN55212-2 on cellular viability, colony formation and 3D tumor sphere in HCT-8 cells. Meanwhile, TP-0903 plus WIN55212-2 treatment promoted the apoptosis of HCT-8 cells. We then investigated the synergistic anti-tumor effect of TP-0903 and WIN55212-2 using colon cancer cell xenografts in immunocompromised and immunocompetent mice. The in vivo study demonstrated that combined administration of TP-0903 plus WIN55212-2 effectively reduced tumor volume and microvessel density and promoted apoptotic cells of tumor tissues in HCT-8 exogenous mice compared to either TP-0903 or WIN55212-2 treatment alone. Moreover, in addition to tumor suppression, the combination therapy of TP-0903 and WIN55212-2 induced the infiltration of cytotoxic CD8^+^ T cells and significantly reduced mTOR and STAT3 activation in tumor tissues of C57BL/6J mice bearing MC-38 cells. **Conclusions**: This study demonstrated that targeting AXL could sensitize cannabinoids to cancer therapy by interfering with tumor cells and tumor-infiltrating CD8^+^ T cells.

## 1. Introduction

With the development of comprehensive treatment approaches such as chemoradiotherapy, targeted therapy and immunotherapy followed by surgery, improvements in the prognosis and overall survival of patients with advanced cancer are evident. However, many patients with advanced, incurable cancer suffer from inevitable side effects of cancer and its treatment, such as anorexia, pain, nausea, vomiting, etc. These affect not only the quality of life of patients and their families but also their comorbidity and possibly tumor progression. So, it is important to find ways to prevent the development of these affective disorders during cancer treatment.

In the popular media, many advocate for the use of cannabis in the treatment of cancer-related symptoms. There is a long history of its use for cancer pain and/or pain, nausea and cachexia induced by cancer treatment [1,2]. To date, some cannabinoid-based medications have already been approved in various countries, such as nabilone and dronabinol capsules, Δ9-tetrahydrocannabinol/cannabidiol oromucosal spray or oral solution of cannabidiol [2,3]. Cannabinoids are a class of lipophilic molecules found in the cannabis plant, which can be divided into three main groups: phytocannabinoids, endocannabinoids and synthetic cannabinoids. The multiple effects of cannabinoids and endocannabinoids are mainly mediated by binding toward G protein-coupled receptors (GPCRs): cannabinoid receptor 1 (CB_1_R) and cannabinoid receptor 2 (CB_2_R). CB_1_R is expressed at high levels in the central nervous system (CNS) where it regulates the synaptic transmission and psychoactive effect [4]. CB1R was also found in peripheral tissues, including skeletal muscle, the liver, the pancreas and adipose tissue [5]. CB_2_R was mainly expressed in immune cells, such as myeloid, macrophage, erythroid, lymphoid and mast cells [6]. Activation of CB_1_R or CB_2_R leads to an inhibition of adenylyl cyclase via G-proteins (Gi/o), which in turn activates many metabolic pathways, including mitogen-activated protein kinases (MAPKs), phosphoinositide 3-kinase (PI3K), protein kinase B (Akt)/mTORC1, the cyclooxygenase-2 pathway (COX-2), ion channels, etc. [7,8]. In recent years, cannabinoids or synthetic cannabinoids such as WIN55-212, JWH-133 and JWH-015 have been extensively studied for their potential anti-cancer effects. In vitro and in vivo cancer models showed that cannabinoids are able to regulate essential cellular processes involved in the inhibition of tumor cells’ proliferation, migration and angiogenesis, as well as the induction of apoptosis and autophagy [9,10,11,12,13]. In view of this, the evaluation of cannabinoids on tumor therapy has become an area of considerable interest to the medical and research communities.

Combination strategies have been a standard approach for cancer treatment. High-throughput RNAi (HT-RNAi) screening is an efficient technique commonly used for the discovery of new drug targets and the identification of essential or synthetic lethal genes that confer resistance or sensitivity to drug treatment [14]. Here, we used a human kinome-wide siRNA library targeting a total of 675 genes involved in crucial cancer pathways to identify the targeted genes that sensitize the anti-proliferation of the cannabinoid agonist WIN55212-2. Among these target genes, AXL is the most promising candidate. AXL is a member of the Tyro3-AXL-Mer (TAM) receptor tyrosine kinases (RTK), which are frequently overexpressed in multiple tumors [15,16,17]. In addition, AXL has been reported to be associated with the acquisition of drug resistance and poor prognosis [18,19,20,21,22]. AXL inhibition in cancer has been shown to induce cell apoptosis and DNA damage, decrease cell proliferation and migration, and improve sensitivity to chemotherapeutic drugs [23,24,25,26]. Recently, increasing evidence suggests that AXL activation shapes an immunosuppressive tumor microenvironment [27,28]. Therefore, it is meaningful to evaluate the synergistic anti-tumor effect of AXL inhibition with WIN55212-2 in vitro and in vivo, and uncover its underlying mechanism.

In this study, we demonstrated targeting AXL as a synergistic treatment in combination with WIN55212-2 by a significant reduction in cellular viability, colony formation and 3D tumor spheres, along with induction of the apoptosis of HCT-8 colon cells. The synergistic anti-tumor effect of TP-0903 and WIN55-212 in vivo was attributed to the promotion of apoptotic cells and reduction in microvessel density in tumor tissue, but also to an increase in CD8^+^ cytotoxic T cells infiltrating into the tumor microenvironment. This study provides the potential combined therapeutic strategy for cancer treatment.

## 2. Results

### 2.1. siRNA Screening Identifies Genes That Sensitizes the Anti-Proliferation of WIN 55212-2

To identify the genes that selectively synergistically inhibit cell proliferation with the cannabinoid receptor agonist WIN55212-2, we screened a panel of siRNAs targeting 675 genes from the human kinome in HepG2 cell lines. Cellular viability was determined after 48 h of reverse transfection of targeting siRNAs or in combination with 5 μM of WIN55212-2 by counting live cells using an INCELL Analyzer 2000, and the results were normalized to RISC-free siRNA (non-targeting siRNA) as a control. Figure 1A shows a ranking of individual siRNAs with or without WIN55212-2 effects on cell viability relative to the control. Subsequently, we calculated the difference in cell proliferation rate between the single siRNA group and the targeted siRNA plus WIN55212-2 group to evaluate their synergistic activity (SA) (Figure 1B). The 49 special siRNAs with SA > 0.09 were screened in Appendix A. Then, 6 of 49 genes, TNK1, AXL, MKNK1, PRKCi, PDGFRL and MINK1, which are closely related to tumorigenesis, were selected and silenced to evaluate the synergistic antiproliferation with WIN55212-2 in cancer cell lines. As shown in Figure 1C,D, silencing of these single genes plus WIN55212-2 treatment could inhibit the proliferation of HCT-8 and HepG2 cells, compared to the silencing targeted RNA alone. The combination treatment of siRNA targeting AXL plus WIN55212-2 showed the most potent anti-proliferation activity.

### 2.2. AXL Inhibitor Effectively Inhibited Tumor Growth in the Presence of WIN55212-2

To confirm the above findings, we used the AXL-specific small molecule inhibitor TP-0903 to evaluate the cell cytotoxicity of AXL inhibition plus WIN55212-2 treatment in multiple cancer cell lines. As shown in Figure 2, TP-0903 effectively decreased the cell viability of NCI-H460, HepG2, BxPC3 and HCT-8 cells in a dose-dependent manner. Treatment with TP-0903 at concentrations ranging from 1 nM to 100 nM plus WIN55212-2 showed potentially synergistic inhibition of proliferation in HCT-8 and Caco-2 cells (synergic score ≥ 10, *p*-value ≤ 0.001) (Appendix A). In addition, we detected the colony formation of TP-0903 treatment alone or in combination with WIN55212-2 in HCT-8 cells. Exposure of TP-0903 (0.67 nM) significantly reduced the number of colonies in HCT-8 cells. TP-0903 (0.2 or 0.67 nM) plus WIN55212-2 treatment further significantly inhibited the formation of cell colonies compared to each of the single groups (Figure 3A,B). In the 3D culture assay, HCT-8 cells could be induced to form spheroids. TP-0903 treatment dose-dependently decreased the resazurin fluorescence in the spheroids. Combined treatment with TP-0903 and WIN55212-2 further reduced the fluorescence in spheroids compared to TP-0903 alone (Figure 3C,D). Furthermore, Synergyfinder 3.0 website analysis demonstrated the synergistic inhibition role of TP-0903 plus WIN55212-2 in the formation of spheroids (Appendix A).

### 2.3. Combination of AXL Inhibitor and WIN55212-2 Treatment Promoted the Cell Apoptosis

Furthermore, we evaluated the induction of cell cycle arrest and apoptosis after treatment with the TP-0903 with or without WIN55212-2. Flow cytometry analysis showed that TP-0903 treatment induced the G2/M arrest, whereas WIN55212-2 treatment slightly increased the percentage of cells in the G0/G1 phase. Combined administration of TP-0903 and WIN55212-2 regulated the balance between the G2/M phase and the G0/G1 phase without affecting the cells in the S phase (Figure 4A,C). Annexin V-FITC/ PI double staining assay revealed that the combined applications of TP-0903 (100 nM) and WIN55212-2 significantly increased the percentage of apoptotic cells compared to that of the single-drug group (Figure 4B,D). Western blot assay showed that the TP-0903 plus WIN55212-2 treatment significantly downregulated the expression of the anti-apoptotic protein BCL2 (Appendix A).

### 2.4. Synergistic Anti-Tumor Effect of AXL Inhibitor and WIN55212-2 in the HCT-8 Cell Xenograft Mice

Given the synergistic anti-proliferative effect of AXL inhibition and WIN55212-2 in vitro, its efficacy was further evaluated in the HCT-8 cell xenograft mice. As shown in Figure 5A,B, TP-0903 treatment displayed inhibition of tumor volume in the HCT-8 cell xenografts. The T/C value was 44.50% in the TP-0903 (0.13 mg/kg) plus WIN55212-2 (2.5 mg/kg) group, which was significantly difference compared to the TP-0903 or WIN55212-2 treatment alone groups (*p* ≤ 0.05). Similarly, the tumor mass was significantly reduced by 49.48% in the 0.13 mg/kg of TP-0903 plus WIN55212-2 group (Figure 5C). And, TP-0903 treatment alone or in combination with WIN55212-2 did not affect the body weight of HCT-8 cell xenograft mice (Appendix A). Additionally, TUNEL staining data showed that colon tissues exhibited obvious apoptosis after treatment with TP-0903 or WIN55212-2. The amounts of apoptotic cells were significantly increased after exposure to a high dose of TP-0903 plus WIN55212-2, compared with those of the single-treated group (*p* ≤ 0.01; *p* ≤ 0.001) (Figure 5D,E). CD31 staining data showed that a high dose of TP-0903 significantly decreased the MVD in tumor tissues. Treatment with TP-0903 (0.13 mg/kg) plus WIN55212-2 showed a further reduction in CD31-positive cells in sections (Figure 5F,G).

### 2.5. Synergistic Anti-Tumor Effect of AXL Inhibitor and WIN55212-2 in the MC38 Cell Xenograft Mice

In addition, we detected the tumor growth inhibition of TP-0903 plus WIN55212-2 using a murine colon MC38 cell xenograft model. As shown in Figure 6A, 50 mg/kg of TP-0903 remarkably inhibited the tumor growth of MC38 cell-bearing mice, with a tumor inhibition rate of 49.29% (*p* ≤ 0.05). Compared to the WIN55212-2 group, TP-0903 plus WIN55212-2 treatment significantly inhibited tumor growth (*p* ≤ 0.01). Flow cytometry analysis was performed to investigate the effects of T lymphocyte infiltration in tumor tissues after administration of TP-0903 plus WIN55212-2. As shown in Figure 6B, TP-0903 treatment did not affect the percentage of CD3-positive cells in CD45^+^ cells infiltrating the tumor tissue, but TP-0903 plus WIN55212-2 treatment significantly increased the proportion of CD3^+^ cells compared to the control group (*p* ≤ 0.05). Co-immunostaining of CD4, CD8, CD3 and CD45 showed that TP-0903 treatment alone or in combination with WIN55212-2 significantly increased the amount of cytotoxic CD8^+^ T lymphocytes (CTLs) in tumor tissue, whereas it significantly decreased the percentages of CD4^+^ helper T lymphocytes (Th) infiltrating the tumor tissue (Figure 6C). Given the role of AXL in activating the downstream AKT/mTOR and STAT3 signaling pathways, we investigated the effects of TP-0903 plus WIN55212-2 treatment on the protein expressions of key molecules by Western blotting assay. As shown in Figure 6D, compared to TP-0903 or WIN55212-2 treatment alone, TP-0903 plus WIN55212-2 treatment significantly suppressed the p(S2448)-mTOR and p(Y705)-STAT3 expressions in tumor tissues of MC38 cell-bearing mice.

## 3. Discussion

Currently, combination therapy is generally becoming a feasible treatment for clinical cancer patients, which could improve drug efficacy, overcome chemoresistance and reduce side effects. In this study, we used a targeted RNAi screen to identify AXL as a potential target that sensitizes the anti-proliferation of the cannabinoid agonist WIN55212-2.

As a member of the receptor tyrosine kinase family, AXL has a well-characterized oncogenic role in proliferation, metastasis and chemoresistance via the activation of pathways such as PI3K/AKT, NK-κB, MAPK, JAK-STAT, etc. [22,29,30]. AXL was overexpressed in a wide range of cancers including breast, lung and ovarian cancer, and the expression level was directly correlated with poor prognosis of cancer [30,31,32]. In early-stage CRC tissues, the expression of AXL was reported as a prognostic biomarker for poor overall survival (OS) [33]. TP-0903 is a small-molecule oral AXL inhibitor that is currently undergoing evaluation in a first-in-human study in patients with advanced solid tumors (CinicalTrials.gov; NCT02729298). The synthetic cannabinoid WIN 55212-2 exhibited anti-tumor effects, in addition to its known palliative effects on some cancer-related symptoms. However, the synergistic anti-tumor effect of TP-0903 and WIN 55212-2 has not been elucidated. Our study found that TP-0903 could induce G2/M phase arrest while WIN 55212-2 mainly induced G0/G1 arrest to inhibit tumor cell growth, as previously reported [34]. The combined anti-tumor effect of TP-0903 and WIN 55212-2 was attributed to the enhancement of apoptotic cancer cells in vitro. The results from the TUNEL assay in vivo further validated their synergistic anti-cancer effect. Furthermore, combination therapy with TP-0903 plus WIN 55212-2 could effectively reduce the CD31-positive cells, suggesting their synergistic anti-angiogenesis effect.

In addition to direct effects on tumor cells, AXL and cannabinoids have been reported to regulate the immune cells. In some cancers, AXL appears to promote the immunosuppressive tumor microenvironment [35,36]. Pharmacological inhibition and genetic knockout of AXL decreased surface expression of MHC-I or the immune checkpoint ligand PD-L1 and increased the secretion of immunosuppressive chemokines [37,38,39]. The cannabinoid receptor is commonly expressed in immune cells. Previous studies have shown that cannabinoids can affect the number and proliferation of T and B cells, but may also have important effects on cytokine secretion or Ig production [40]. However, the effect of cannabinoids on T cells infiltrating the tumor tissue has not been investigated. In our study, we also investigated the anti-tumor effect of combined treatment with TP-0903 and WIN55212-2 in immunocompetent mice. Here, we found that treatment with TP-0903 plus WIN55212-2 could increase the number of T lymphocytes infiltrating the tumor tissue, especially the cytotoxic CD8^+^ T cells. Taken together, we speculated that the synergetic anti-tumor effect of TP-0903 plus WIN55212-2 treatment may be attributed not only to their role in anti-proliferation and induction of apoptosis, but also to the induction of cytotoxic T lymphocytes killing cancer in the tumor microenvironment.

## 4. Materials and Methods

### 4.1. Cell Lines and Culture

The human colon cancer cells (HCT-8, HCT-29 and Caco-2), lung cancer cells (NCI-H446, NCI-H460 and NCI-H462), hepatocellular carcinoma cells (HepG2, PLC), prostate cancer cells (PC3), breast cancer cells (MDA-MB-231), pancreatic cancer cells (BxPC3) and ovarian adenocarcinoma cells (SKOV3) were obtained from the American Type Culture Collection (ATCC, Manassas, VA, USA) or the Cell Culture Center at the Chinese Academy of Sciences. These cells were cultured in DMEM or RPMI-1640 containing 10% fetal bovine serum (FBS), streptomycin 100 µg/mL and penicillin 100 U/mL at 37 °C in humidified 5% CO_2_.

### 4.2. Chemicals and Antibodies

WIN55212-2 (Cat. W102) was purchased from Sigma (St. Louis, MO, USA); AXL inhibitor TP-0903 (Cat. S7846) was obtained from Selleck. Resazurin (Cat. R8150) was purchased from Solarbio. CellLight^TM^ Histone 2B-GFP (Cat. C10594) and Hoechst33342 (Cat. H3570) were purchased from ThermoFisher (Waltham, MA, USA). Lipofectamine^TM^ RNAi MAX (Cat. 13778150) was purchased from Invitrogen. Antibodies to phosphor-AXL_Tyr698_, phosphor-AXL_Tyr702_, AXL, phosphor-AKTser473, AKT, phosphor-mTOR_Ser2448_, mTOR, phosphor-STAT3_Tyr705_, STAT3 and β-actin were purchased from Cell Signaling Technologies (Danvers, MA, USA).

### 4.3. siRNA Kinome Library Screen

HepG2 cells were screened for siRNA libraries targeting a total of 675 human kinome genes involved in crucial cancer pathways (siRNA library, Dharmacon, Lafayette, CO, USA). In brief, 3 siRNA duplexes of every single gene were premixed and coated on a cell chip in quadruplicate as previously reported [41]; HepG2 cells were then seeded into the chip as approximately 60% confluent. After 24 h, the control plate was added to the vehicle (0.1%DMSO), while the parallel plate was treated with 5 μM of WIN55212-2 for an additional 48 h. Then, CellLight^TM^ Histone 2B-GFP was added to label the live cell. The cell images were visualized using an INCELL Analyzer 2000, and the cell counts were analyzed by CellProfiler software (Version 3.0) tools. The relative proliferation of different siRNA-treated cells was normalized by the proliferation rate of non-targeting siRNA. The synergistic activity (SA) was calculated as follows: Relative Proliferation _siRNA_-Relative Proliferation _siRNA+WIN55212-2_. The genes that did not almost affect cell proliferation after silencing themselves (|relative proliferation rate| < 1.09), with SA being more than 0.09, were identified as target genes in our study. For validation, we selected 6 genes that participated in the tumorigenesis to detect the cell proliferation after silencing these targeted genes and in combination with WIN55212-2 treatment in cancer cell lines using the MTT assay.

### 4.4. Antiproliferation Assay

The MTT assay was used to validate the cellular viability inhibition of AXL inhibitor TP-0903 plus WIN55212-2 treatment in multiple cancer cell lines. Briefly, cancer cells were seeded into 96-well plates overnight, followed by administration with indicated concentrations of TP-0903 or in combination with 5 μM of WIN55212-2 for another 48 h. Then, MTT solution (0.5 mg/mL) was added and incubated for 4 h at 37 °C. Then, the absorbance of formed formazan crystals solution was measured at 550 nm using a microplate reader (Biotek Instruments, Winooski, VT, USA).

For the colony formation assay, HCT-8 cells were seeded at a density of 800 cells/well into 12-well plates. After 24 h, cells were treated with TP-0903 (0.2 nM and 0.667 nM) alone or in combination with 0.333 μM of WIN55212-2 until the formation of cell colonies. Then, cells were fixed with 4% paraformaldehyde (P1110, Solarbio, Beijing, China) for 10 min, and stained with methyl violet (C8470, Solarbio, Beijing, China) for another 10 min. Digital images of colonies were recorded using Zeiss axio observer (Carl Zeiss, Oberkochen, Germany) and counted using Image J software (Version: 1.53K).

For the 3D tumorsphere culture assay, HCT-8 cells (1 × 10^3^ cells/well) were seeded into agar (2%)-coated 96-well plates. The supernatant was renewed with half fresh medium every two days. When sphere sizes reached 100–200 μM, TP-0903 (1, 3, 10, 30, 100 nM) was added alone or combined with 5 μM of WIN55212-2 for 72 h. After adding 0.01% resazurin for an additional 6 h, the fluorescent intensity of the spheres was measured using a microplate reader (SpectraMax M5/M5e, Ex/Em: 560/590 nM). The fluorescent images were collected using the ImageXpress Micro high-content screening system at the 10× objective lens and the sizes of the tumor spheres were analyzed.

### 4.5. Cell Cycle and Apoptosis Assay by Flow Cytometry

HCT-8 cells were cultured in the presence of TP-0903 (30 or 100 nM) alone or in combination with 5 μM of WIN55212-2 for 48 h. Cells were then harvested and fixed with 70% ice-cold ethanol overnight at −20 °C. To determine DNA content, cells were stained with FxCycle™ PI/RNase Staining Solution (No. F10797, Invitrogen, Carlsbad, CA, USA) for 30 min, and finally analyzed by BD FACS flow cytometry. For the apoptosis assay, cells were resuspended in 100 μL of staining buffer and then stained with 5 μL of Annexin V- FITC and PI for 15 min in the dark, and the apoptotic cells were analyzed using flow cytometry.

### 4.6. Western Blotting Assay

Treated samples were lysed in RIPA buffer (50 mM Tris, 150 mM NaCl, 1% Triton X-100, 1% sodium deoxycholate, 0.1% SDS, 1 mM EDTA, pH 7.4) including 1% protease inhibitor cocktails (Sigma-Aldrich Co., Saint Louis, MO, USA) for 30 min on ice. The lysates were then centrifuged at 13,000 rpm for 10 min and the supernatants were collected for BCA assay. Equal protein samples (60 µg) were subjected to Western blotting on SDS-PAGE, immunoblotted with appropriate antibodies (1:1000) and visualized using Tanon ™ High-sig ECL Western Blotting Substrate (Tanon, China). Signals were detected by Image Quant LAS 4000 (GE Healthcare, Piscataway, NJ, USA). Anti-β-actin antibody was used as a loading control.

### 4.7. Mouse Xenograft Study

All animal experiments were approved by the Ethics Committee for Animal Experiments and were performed in compliance with the regulations and guidelines of the Institute of Materia Medica, Chinese Academy of Medical Sciences & Peking Union Medical College (approval number: 00009626). BALB/c nude mice and C57BL/6J mice (18–20 g) were purchased from the Institute of Experimental Animal Research, Peking Union Medical College (Beijing, China). HCT-8 cells (2 × 10^6^/200 μL PBS) were subcutaneously injected into the flank of BALB/c nude mice. After the tumor volume reached 1500~2000 mm^3^, tumor tissues were chipped into 2 × 2 × 2 mm^3^ pieces and transplanted to the left franks of the nude mice. When the transplanted tumor grew to an average of 100 mm^3^, the mice were randomly divided into 7 groups: mice intratumorally injected with TP-0903 (3.85 μg/kg or 0.13 mg/kg, QD), mice treated with WIN55212-2 (2.5 mg/kg, i.p.) every two days, mice treated with WIN55212-2+ TP-0903 (3.85 μg/kg or 0.13 mg/kg), mice treated with Cisplatin (5 mg/kg, i.p., once a week) and mice treated with vehicle. The body weight, tumor volume and tumor weight of mice were recorded. Tumor volume was calculated as length × width^2^ × 0.5.

MC38 cells (2.5 × 10^5^/200 μL PBS) were injected subcutaneously into the right flank of the C57BL/6J mice. The following day after tumor implantation, mice were randomly distributed into 5 groups: vehicle group, TP-0903 (50 mg/kg, p.o.) group, WIN55212-2 (2.5 mg/kg, i.p.) group, TP-0903 plus WIN55212-2 group and CTX group (80 mg/kg, i.p.). At the end of the experiments, mice were sacrificed and the tumor tissues were weighed, and then fixed or digested for further experiments.

### 4.8. Tumor Infiltrating Lymphocytes (TIL) Analysis by Flow Cytometry

For the analysis of lymphocyte cell infiltration in xenograft tumor tissue, the tumor was minced into small pieces and incubated with 1 mg/mL collagenase Type IV plus 0.1 mg/mL DNaseI (Cat. #C5138 and #D5025, Sigma Aldrich, Saint Louis, MO, USA) at 37 °C for 30 min. The cell suspension was washed and filtered through a 70 μm cell strainer (BD Biosciences, Bedford, MA, USA), then stained with PE-conjugated anti-mouse CD45 antibody, Alexa Flour 488 anti-mouse-CD3 antibody, allophycocyanin (APC)/Cy7-conjugated anti-mouse CD4 antibody and PerCP-Cy5.5-conjugated anti-mouse CD8 antibody. The stained cell samples were analyzed using flow cytometry (BD FACSVerse). The proportion of positively stained cells was analyzed with the FlowJo software package (Tree Star, Ashland, OR, USA).

### 4.9. TUNEL Assay

Tumor tissue apoptosis was determined using a colorimetric TUNEL Apoptosis Assay Kit (C1091, Beyotime, Shanghai, China). In brief, tissue sections were dewaxed, dehydrated and treated with proteinase K, followed by incubation with 50 μL of TUNEL reaction buffer containing 5 μL of TdT enzyme and 45 μL of biotin-labeled dUTP at 37 °C for 1 h. The reaction was stopped and the sections were washed with PBS (LVN10022, Livning, Beijing, China). Then, 50 μL of Streptavidin-HRP working solution was added for 30 min. After washing, the apoptotic cells were visualized with DAB solution (zsbio, Beijing, China). The cell nuclear was then stained with hematoxylin. The sections were scanned by the microscope. The TUNEL-positive rate was calculated by Image-Pro plus 5.1 software.

### 4.10. Immunohistochemical (IHC) Staining Assay

Microvessel density (MVD) was assessed by immunostaining for CD31. Sections (4 µm thick) were de-paraffinized and rehydrated followed by antigen retrieval in citrate solution (P0081, Beyotime, Shanghai, China). After incubated in 3% H_2_O_2_ (PV-9000, ZSBIO, Beijing, China) to inactivate endogenous peroxidase activity, the section was blocked using 3% BSA (0332, Lablead, Beijing, China) at room temperature for 30 min, and incubated with mouse monoclonal anti-CD31 antibody (Cat. 3528; CST) at 4 °C overnight. CD31-stained cells were visualized with a DAB kit after incubating with horseradish peroxidase (HRP)-labeled antibody. The cell nuclear was labeled with hematoxylin for 3 min. Then, the sections were sealed with a mounting buffer and examined using a microscope. Three pictures were taken randomly, and the density of brown-colored microvessels was evaluated by Image-Pro plus 5.1 software.

### 4.11. Statistical Analysis

The data are expressed as the means ± standard deviation (S.D). ANOVA followed by LSD and a two-sided Student’s t-test were applied to analyze the experimental differences. A *p*-value of <0.05 was considered to be statistically significant.

## 5. Conclusions

In summary, we identify that AXL inhibition plus WIN55212-2 treatment exhibited synergistic anti-tumor effects in vitro and in vivo. AXL inhibition potentiated the inhibitory role of WIN55212-2 on cell viability, colony formation and 3D tumor sphere, as well as the induction of apoptosis in HCT-8 cells. In addition, the combination of AXL inhibitor with WIN55212-2 treatment exhibited an increase in cytotoxic T lymphocytes, leading to cancer death. These findings pointed out that, based on RNAi screen technology, targeting AXL could be a potential therapeutic strategy for enhancing cannabinoid efficacy in colon cancer.

## Figures and Tables

**Figure 1 pharmaceuticals-17-01465-f001:**
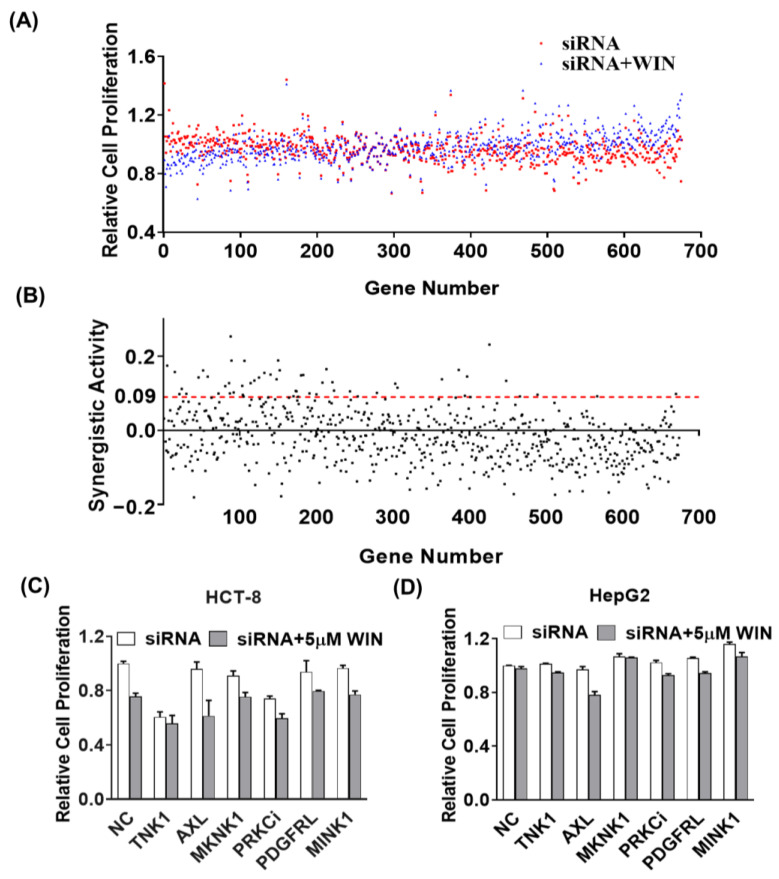
Identification of kinases that synergistically inhibit cancer cell viability with WIN55212-2 using siRNA screening technology. The siRNA libraries targeting a total of 675 human kinome genes were coated onto a cell chip. After seeding with the HepG2 cells for 24 h, the plate was added with the vehicle (0.1% DMSO) or treated with 5 μM of WIN55212-2 for an additional 48 h. The counts of live cells were calculated by staining with CellLight^TM^ Histone 2B-GFP and visualized using an INCELL Analyzer 2000 (GE Healthcare, Parsippany, NJ, USA). (**A**) Relative cell proliferation of individual siRNA with or without WIN55212-2 in HepG2 cells. (**B**) The synergistic activity (SA) was calculated by the difference in cell proliferation rates between the single siRNA group and the targeted siRNA plus WIN55212-2 group. The relative cell proliferations of silencing six targeted siRNAs with or without WIN55212-2 in HCT-8 cells (**C**) and HepG2 cells (**D**).

**Figure 2 pharmaceuticals-17-01465-f002:**
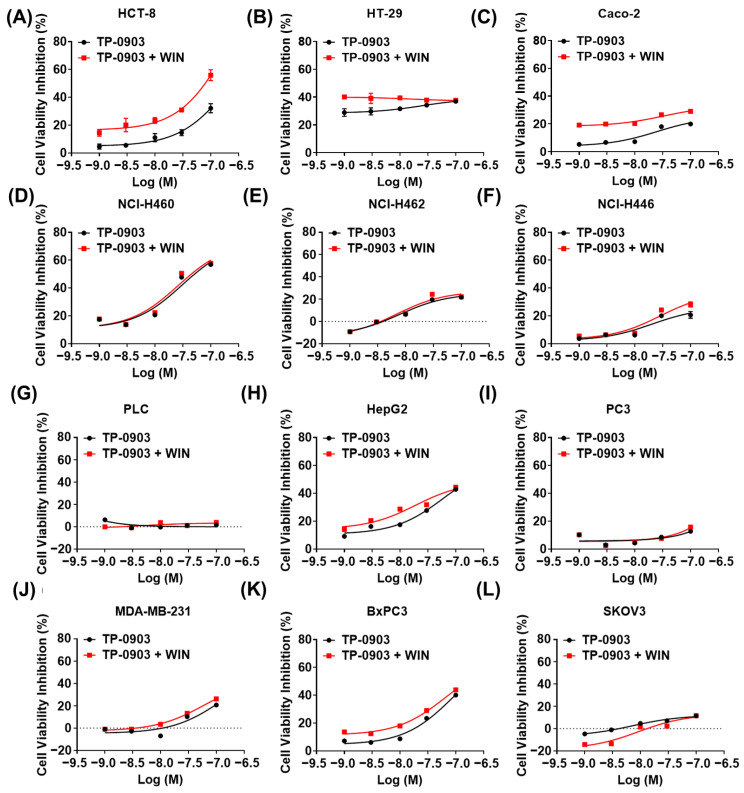
The anti-proliferative role of AXL inhibitor combined with WIN55212-2 in cancer cells. (**A**–**L**) Several cancer cell lines were treated with different doses of the AXL-specific small molecule inhibitor TP-0903 alone or in combination with 5 μM of WIN55212-2 for 48 h; the inhibition of cell viability was detected by MTT assay.

**Figure 3 pharmaceuticals-17-01465-f003:**
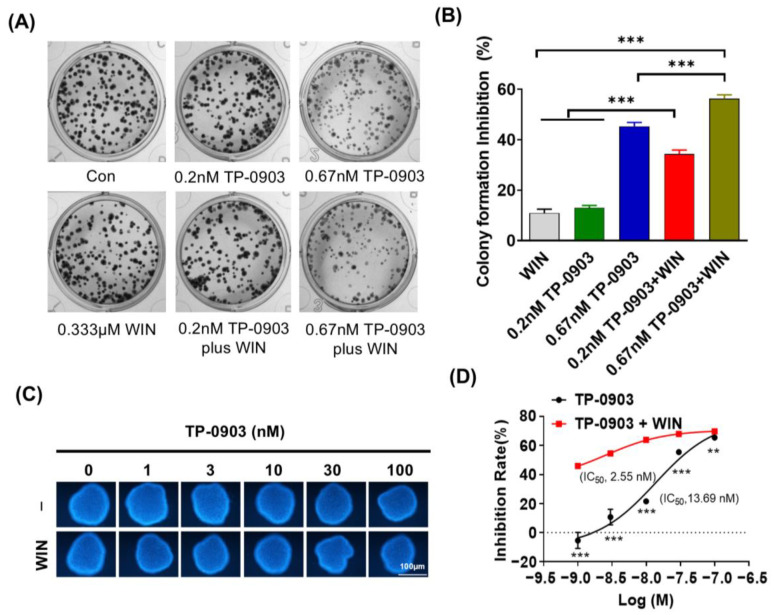
The colony- and spheroid-forming ability of AXL inhibitor plus WIN55212-2 treatment. (**A**) HCT-8 cells were treated with TP-0903 (0.2 nM or 0.667 nM) alone or in combination with 0.333 μM of WIN55212-2 for 10 days; the cell colonies were stained with methyl violet and photographed. (**B**) The inhibition of cell colony formation was calculated from three experiments and is shown in the histogram. (**C**) HCT-8 cells were seeded into agar (2%)-coated 96-well plates until the size of the spheres reached 100–200 μM. Then, the indicated doses of TP-0903 (1, 3, 10, 30, 100 nM) were added alone or in combination with 5 μM of WIN55212-2 for 72 h. The fluorescence intensity of the sphere was detected after staining with 0.01% resazurin using a microplate reader. (**D**) The inhibition of cell spheroid formation was calculated from three experiments and is shown in the histogram. ** *p* < 0.01, *** *p* < 0.001 indicate a significant difference (TP-0903/WIN treat alone vs. Combination therapy).

**Figure 4 pharmaceuticals-17-01465-f004:**
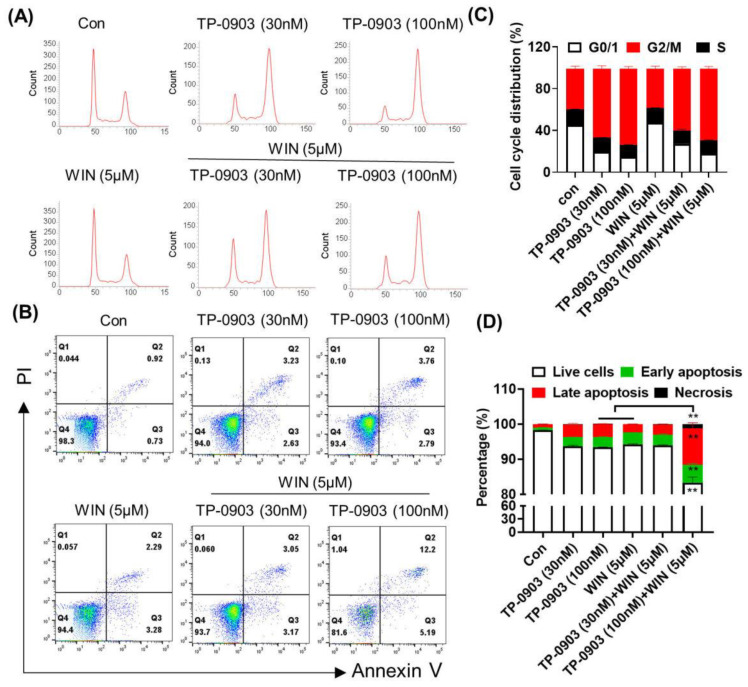
The effect of AXL inhibitor plus WIN55212-2 treatment on cell cycle arrest and apoptosis in HCT-8 cells. HCT-8 cells were treated with TP-0903 (30 nM or 100 nM) alone or in combination with 5 μM of WIN55212-2 for 24 h or 48 h and subjected to flow cytometry to determine cell cycle arrest (**A**) and apoptosis (**B**). (**C**) The percentages of the cells in G0/1, G2/M and S phases are shown in the histogram. (**D**) The percentages of live cells, early-stage apoptosis, late-stage apoptosis and necrotic cells are shown on the quadrants. Data are expressed as the mean ± SD from three independent experiments. ** *p* < 0.01 indicate a significant difference (TP-0903/WIN treat alone vs. Combination therapy).

**Figure 5 pharmaceuticals-17-01465-f005:**
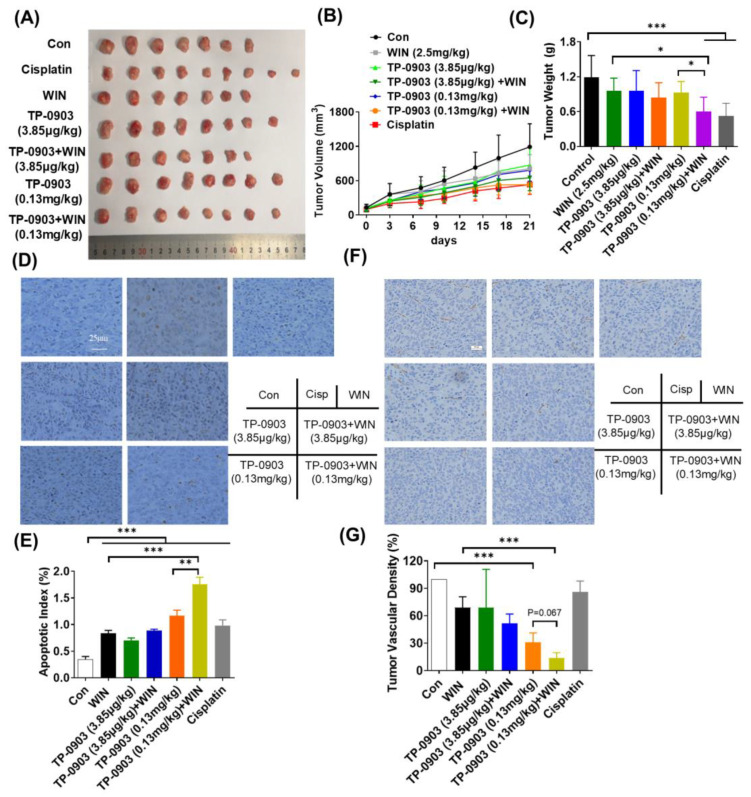
The anti-tumor activity of AXL inhibitor plus WIN55212-2 treatment in the HCT-8 cell xenograft mice. BALB/c nude mice were injected subcutaneously in the flank with HCT-8 colon cancer pieces and intratumorally treated with TP-0903 (3.85 μg/kg or 0.13 mg/kg) alone or in combination with WIN55212-2 (2.5 mg/kg) every two days. Cisplatin was used as a control (5 mg/kg per week). The tumor photograph (**A**), tumor volume (**B**) and tumor mass (**C**) of individual mice were recorded. (**D**) The apoptosis of tumor sections was determined by a colorimetric TUNEL assay. (**E**) The TUNEL-positive rate was calculated and shown in the histogram. (**F**) The microvessel density of tumor tissue was evaluated by immunostaining. (**G**) The percentage of CD31-positive cells was examined and is shown in the histogram. * *p* < 0.05, ** *p* < 0.01, *** *p* < 0.001 indicate a significant difference.

**Figure 6 pharmaceuticals-17-01465-f006:**
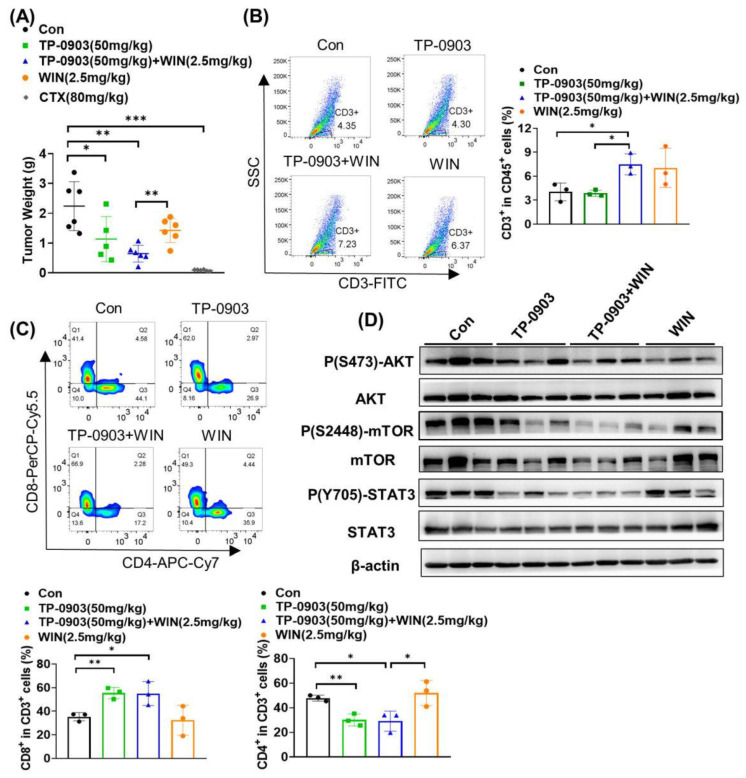
The anti-tumor activity of AXL inhibitor plus WIN55212-2 treatment in the MC38 cell xenograft mice. C57BL/6J mice were injected subcutaneously in the right flank with MC38 colon cancer cells at a density of 2.5 × 10^5^ and treated with 50 mg/kg TP-0903 alone or in combination with WIN55212-2 (2.5 mg/kg, i.p.) every two days. CTX (80 mg/kg) was injected once as a control group. (**A**) The tumor weight of individual mice was detected at the end of the experiments. The infiltration of T lymphocytes in the tumor tissues was evaluated by flow cytometry. The percentages of CD3^+^ cells in CD45^+^ cells (**B**), CD8^+^ cells in CD3^+^ cells and CD4^+^ cells in CD3^+^ cells (**C**) infiltrating tumor sections were shown in the diagram. (**D**) The expressions of Ser473-phosphorylated AKT (p-AKT_Ser473_), Ser2448-phosphorylated mTOR (p-mTOR_Ser2448_), Tyr705-phosphorylated STAT3 (p-STAT3_Tyr705_), AKT, mTOR and STAT3 in tumor tissues were evaluated by Western blot analysis. * *p* < 0.05, ** *p* < 0.01, *** *p* < 0.001 indicate a significant difference.

## Data Availability

Data is contained within the article, further inquiries can be directed to the corresponding author.

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
