# Peer review of "RNAi Screen Identifies AXL Inhibition Combined with Cannabinoid WIN55212-2 as a Potential Strategy for Cancer Treatment"

_pharmaceuticals, 2024, doi:10.3390/ph17111465_

Round 1
Reviewer 1 Report
Comments and Suggestions for Authors
The article by Li et al. identifies AXL as a new therapeutic target for cancer. A specific compound, TP-0903, can be used to inhibit AXL and combined with the cannabinoid WIN55212 in cancer treatment. The document is well structured and written. However, the combination's effect is not very strong.
In Figure 2, the authors considered 12 different tumor lines. Only some of these lines are sensitive to the WIN/TP combination. Do the cells considered in the paper express the protein encoded by AXL? Are the differences in the figure statistically significant?
The authors describe the effect of the combination of TP-0903 and WIN as synergistic. However, to ensure the accuracy of this claim, it is crucial that specific tests, such as those using software and methods like the Bliss Independence Model or Isobolographic Analysis, are carried out.
In Figure 3. How is the inhibition of colony formation calculated? To carry out the experiments described in this figure, the authors used lower drug concentrations (WIN0.333uM) compared to the 5uM used for the viability assays. They should comment on this point.
Figure 5 in panel C would be better to show the tumour weight rather than the tumour weight inhibition in the graph, a parameter derived and not obtained from direct measurements.
Comments on the Quality of English LanguageMinor editing of English language required.
line 91 Apoptosisi
Author Response
Dear Editor and Reviewer:
We are honored to get your support and appreciation of our work! Thank you very much for these precious comments concerning my manuscript entitled “RNAi Screen identifies AXL inhibition combined with cannabinoid WIN55212-2 as a potential strategy for cancer treatment” (pharmaceuticals-3232234).
These comments are all valuable and very helpful for revising and improving my paper, as well as the important guiding significance to my researches. We have studied comments carefully and have addressed each point by reviewer below.
We hope meet with approval!
Best Regards!
Yours sincerely,
Nina Xue, Ph.D., Associated Professor
Responds to the reviewer’s comments:
Reviewer #1:
- In Figure 2, the authors considered 12 different tumor lines. Only some of these lines are sensitive to the WIN/TP combination. Do the cells considered in the paper express the protein encoded by AXL? Are the differences in the figure statistically significant?
Response: Thank you for your suggestion! By detecting the AXL expressions using the CCLE database, we found that there was no significant correlation between drug sensitivity and AXL expression in these cell lines. As a member of receptor tyrosine kinase family, AXL is activated by various factors (growth factors, cytokines, and extracellular matrix components) to promote the tumor progress by upregulating the multiple downstream signaling pathways including PI3K-AKT, MAPK, JAK-STAT, SRC-FAK, etc. The factors that determine the drug sensitivity are numerous and complex, and they also vary depending on the type of cells. In addition to mutation or overexpression of gene encoding drug target, the activation of its specific signaling pathways also affect the drug sensitivity. In the supplementary Fig.3S, we found that the expressions of p-AXL(Y698), p-STAT3(Y705) and anti-apoptosis marker Bcl2 were significantly decreased after treatment with TP-0903 in combination with WIN, further demonstrating their potential synergistic anti-tumor effect in HCT-8 cells.
Synergic score is a known index to evaluate the combined effect of multiple drugs or treatments. We calculated the synergic score of WIN/TP combination in these different 12 cancer cell lines. Only in HCT-8 and Caco-2 cells, the synergic scores of WIN/TP combination are greater than 10, indicating the synergistic anti-tumor effect of TP-0903 plus WIN treatment. We pointed the synergistic antitumor effect in revised manuscript and supplementary data.
- The authors describe the effect of the combination of TP-0903 and WIN as synergistic. However, to ensure the accuracy of this claim, it is crucial that specific tests, such as those using software and methods like the Bliss Independence Model or Isobolographic Analysis, are carried out.
Response: Thank you for your suggestion! The SynergyFinder 3.0 website was used for interactive analysis and visualization of multi-drug combination response data. Multiple synergy reference models (ZIP, HAS, Loewe and Bliss) were used to evaluate the synergistic antitumor effect of TP-0903 and WIN in Figure 3D. The synergy scores in these four models were 12.19, 20.91, 18.13, and 13.07, respectively (p-values ≤ 0.001) (Synergy Score ≥ 10, synergistic) (https://synergyfinder.fimm.fi/synergy/synfin_docs/). The synergy scores were shown as follow and in the revised supplementary data (Fig. S2).
Figure. The Synergy Score of TP0903 plus WIN55212-2 treatment in HCT-8 cells using the ZIP, HAS, Loewe and Bliss models in SynergyFinder 3.0 website analysis.
- In Figure 3. How is the inhibition of colony formation calculated? To carry out the experiments described in this figure, the authors used lower drug concentrations (WIN0.333uM) compared to the 5uM used for the viability assays. They should comment on this point.
Response: The inhibition of colony formation is typically calculated by comparing the number or size of colonies formed in the control group (without the inhibitory agent) to those formed in the treatment group (with the inhibitory agent). The formula for calculating the inhibition rate is as follow: (number of colonies in control group - number of colonies in treatment group) / number of colonies in control group) × 100 %.
The colony formation inhibition and cellular viability inhibition are two complementary measures that provide different insights into the an-tumor effects of drugs or chemical agents on cells. The colony formation assay is an in vitro cell survival assay based on the ability of a single cell to grow into a colony. It is a long-term measure as it requires days to weeks for colonies to form. It reflects the cumulative effects of a treatment on cell proliferation, survival, and the ability to establish new colonies. The cell viability assay is essentially used for screening the proportion of living cells at a specific time point. In general, colony formation inhibition may be more sensitive to lower concentrations or milder treatments that have a subtle but cumulative effect on cell growth over time. Cellular viability inhibition may be more sensitive to higher concentrations or more potent treatments that cause immediate cell death. Therefore, we chose the lower concentrations of WIN (0.333 μM) in the colony formation assay where it exhibited significant inhibition of colonies. In the cellular viability assay, the IC20 value of WIN for 48 h was used for detecting the synergistic antitumor effect.
- Figure 5 in panel C would be better to show the tumour weight rather than the tumour weight inhibition in the graph, a parameter derived and not obtained from direct measurements.
Response: Thank you for your suggestion! The quantification data of tumor weight for indicated group was shown in the revised Figure 5C.

Reviewer 2 Report
Comments and Suggestions for Authors
Comments:
1. The abstract and conclusion should be revised to show and highlight the manuscript's objectives in treating cancer cells. This study aimed to identify the target genes that show a synergistic anti-tumour role in combination with the cannabinoid WIN55212-2 in vitro and in vivo.
2. The introduction should include recent references and be reformulated due to the importance of the biological activity of many drugs, such as AXL inhibition combined with cannabinoid WIN55212-2, for cancer treatment.
3. As a receptor tyrosine kinase family member, AXL has a well-characterized oncogenic role in proliferation, migration, invasion and metastasis. AXL was overexpressed in a wide range of cancers and was directly correlated with poor prognosis of cancer; authors must describe and show in detail in the manuscript (part discussion).
Comments on the Quality of English Language
Minor editing of the English language is required.
Author Response
Dear Editor and Reviewer:
We are honored to get your support and appreciation of our work! Thank you very much for these precious comments concerning my manuscript entitled “RNAi Screen identifies AXL inhibition combined with cannabinoid WIN55212-2 as a potential strategy for cancer treatment” (pharmaceuticals-3232234).
These comments are all valuable and very helpful for revising and improving my paper, as well as the important guiding significance to my researches. We have studied comments carefully and have addressed each point by reviewer below.
We hope meet with approval!
Best Regards!
Yours sincerely,
Nina Xue, Ph.D., Associated Professor
Reviewer #2:
- The abstract and conclusion should be revised to show and highlight the manuscript's objectives in treating cancer cells. This study aimed to identify the target genes that show a synergistic anti-tumour role in combination with the cannabinoid WIN55212-2 in vitro and in vivo.
Response: Thank you for your suggestion! We have highlighted the aim of our research in the abstract and conclusion sections of the revised manuscript.
- The introduction should include recent references and be reformulated due to the importance of the biological activity of many drugs, such as AXL inhibition combined with cannabinoid WIN55212-2, for cancer treatment.
Response: Thank you for your suggestion! We added the recent references and revised the part of introduction.
- As a receptor tyrosine kinase family member, AXL has a well-characterized oncogenic role in proliferation, migration, invasion and metastasis. AXL was overexpressed in a wide range of cancers and was directly correlated with poor prognosis of cancer; authors must describe and show in detail in the manuscript (part discussion).
Response: Thank you for your suggestion! We described the role of AXL in cancer in the introduction section. We added and discussed this content in the part of discussion in the revised manuscript.
